# Whole Genome Sequence Analysis of *Cupriavidus necator* C39, a Multiple Heavy Metal(loid) and Antibiotic Resistant Bacterium Isolated from a Gold/Copper Mine

**DOI:** 10.3390/microorganisms11061518

**Published:** 2023-06-07

**Authors:** Zhenchen Xie, Dan Wang, Ibtissem Ben Fekih, Yanshuang Yu, Yuanping Li, Hend Alwathnani, Martin Herzberg, Christopher Rensing

**Affiliations:** 1Institute of Environmental Microbiology, College of Resource and Environment, Fujian Agriculture and Forestry University, Fuzhou 350002, China; 2College of Chemistry and Environmental Engineering, Shenzhen University, Shenzhen 518060, China; 3Functional and Evolutionary Entomology, Terra, Gembloux Agro-Bio Tech, University of Liege, Passage des Deportes-2, B-5030 Gembloux, Belgium; 4Department of Botany and Microbiology, King Saud University, Riyadh 11495, Saudi Arabia; 5Molecular Microbiology, Institute for Biology/Microbiology, Martin-Luther-University Halle-Wittenberg, Kurt-Mothes-Str. 3, 06120 Halle, Germany

**Keywords:** *Cupriavidus necator*, complete genome, heavy metal resistance, antibiotic resistance, degradation pathways

## Abstract

Here a multiple heavy metal and antibiotic resistant bacterium *Cupriavidus necator* C39 (*C. necator* C39) was isolated from a Gold-Copper mine in Zijin, Fujian, China. *C. necator* C39 was able to tolerate intermediate concentrations of heavy metal(loid)s in Tris Minimal (TMM) Medium (Cu(II) 2 mM, Zn(II) 2 mM, Ni(II) 0.2 mM, Au(III) 70 μM and As(III) 2.5 mM). In addition, high resistance to multiple antibiotics was experimentally observed. Moreover, strain C39 was able to grow on TMM medium containing aromatic compounds such as benzoate, phenol, indole, p-hydroxybenzoic acid or phloroglucinol anhydrous as the sole carbon sources. The complete genome of this strain revealed 2 circular chromosomes and 1 plasmid, and showed the closest type strain is *C. necator* N-1^T^ based on Genome BLAST Distance Phylogeny. The arsenic-resistance (*ars*) cluster *GST-arsR-arsICBR-yciI* and a scattered gene encoding the putative arsenite efflux pump ArsB were identified on the genome of strain C39, which thereby may provide the bacterium a robust capability for arsenic resistance. Genes encoding multidrug resistance efflux pump may confer high antibiotic resistance to strain C39. Key genes encoding functions in degradation pathways of benzene compounds, including benzoate, phenol, benzamide, catechol, 3- or 4-fluorobenzoate, 3- or 4-hydroxybenzoate and 3,4-dihydroxybenzoate, indicated its potential for degrading those benzene compounds.

## 1. Introduction

*Cupriavidus* is a genus of the family *Burkholderiaceae*, members of this genus are well known for their heavy-metal resistance and diverse metabolic capabilities in different niches, especially from heavy metal and organic-chemical contaminated soils, such as halobenzoates, chlorophenols and nitrophenols, and thus making them useful for bioremediation [1,2]. *Cupriavidus necator* (formerly *Wautersia eutropha, Alcaligenes eutrophus* or *Ralstonia eutropha*) is a versatile microorganism found in both soil and water that is able to perform both heterotrophic and chemolithoautotrophic metabolisms depending on environmental conditions [3]. The type strain of this species is a gram-negative, aerobic, mesospheric, short rod that multiplies by binary fission [4]. Some strains of *C. necator* have been of great applied interest for their ability to produce various value-added compounds, such as polyhydroxyalkanoates (PHAs) [5,6], ethanol [7], isobutanol [8], isopropanol [9], methyl ketones [10], 2-hydroxyisobutyric acid [11], and 2-methylcitric acid [12]. Interestingly, the type strain *Cupriavidus necator* N-1 has been shown to be able to attack other bacteria when nutrients in the soil are low [4].

At the time of this writing, there were at least 6 complete genomes of *Cupriavidus necator* strains published, including the type strain *C. necator* N-1 [13], a versatile pollutant degrader *Cupriavidus pinatubonensis* JMP134 [14,15], *C. necator* strain H16 (DSM 428) [16], an azaarene-degrading and polyhydroxyalkanoate-producing *C. necator* strain KK10 [17], 3-chlorobenzoate-degrading *C. necator* strain NH9 [18] and the beta-rhizobial *C. necator* strain UYPR2.512 [19]. In addition, the draft genome of a chlorinated-ethene degrader strain PHE3-6 (NBRC 110655) has been reported previously [20].

Some metal-resistant genera including *Bacillus*, *Arthrobacter*, *Pseudomonas*, *Cupriavidus*, *Stenotrophomonas*, *Desulfovibrio*, *Shewanella*, *Lysinibacillus*, and *Acinetobacter* have been demonstrated to display a high capacity of biosorption or removal of different heavy metals, while other bacteria and archaea belonging to the genera *Acidithiobacillus*, *Leptospirillum*, *Sulfobacillus*, and *Ferroplasma* have mostly been associated with metal minerals and shown to be involved in bioleaching processes [21,22]. Those microbes are of fundamental importance in the bioremediation of metal-contaminated natural habitats and bioleaching of valuable metals from complex minerals.

Here we report the whole genome of another strain of the species *Cupriavidus necator*, namely C39, which was isolated from the soil of a gold-copper mine and characterized by being resistant to multiple heavy metals and antibiotics. Gold is often associated with other heavy metals including arsenic [23]. Therefore, numerous arsenic-resistant bacteria were isolated from gold mines [24]. The soil near gold-copper mines is contaminated by a variety of heavy metals, which selects for improving the tolerance of microorganisms to heavy metals. Our goal was to screen and subsequently select some bacteria with multiple heavy metal resistances, so as to enrich the available resource of described and characterized heavy metal resistant bacteria. Strengthening the research on described strains with multiple heavy metal resistances can lead to a better understanding of the adaptive mechanisms employed by microorganisms in environments polluted by heavy metals. Here, we were able to isolate the arsenic resistant strain *C. necator* C39 containing a unique arsenic resistance determinant. In addition, this strain was shown to contain multiple functional degradation pathways of benzene compounds. Sequencing the complete genome of *C. necator* C39 was intended to expand our understanding of this potentially useful bacterium.

## 2. Materials and Methods

### 2.1. Chemicals

Copper, zinc, nickel, gold and arsenite stock solutions were prepared with CuSO_4_, ZnCl_2_, NiCl_2_, HAuCl_4_ and NaAsO_2_, respectively, and filtered through 0.22 μm micropore membrane for sterilization. The antibiotics were dissolved in their corresponding solvents (streptomycin, gentamycin, kanamycin and ampicillin in double-distilled water; chloramphenicol and tetracycline hydrochloride in absolute ethanol; rifampicin in dimethyl sulfoxide) and filtered through a 0.22 μm micropore membrane.

### 2.2. Isolation of Bacterial Strain C39

The soil sample was taken from the sewage outfall of the Zijin Copper-Gold mine in Fujian, China (GPS: latitude 25°09.708′ N, longitude 116°23.335′ E). Five sampling points were established to collect soil samples at depths of 5 to 10 cm. Inductively coupled plasma mass spectrometry (ICP-MS) was used to determine metal(loid) concentrations in the soil samples. The pH of the soil sample was 3.1, the moisture content was 7%, and the concentrations of various heavy metals were as follows: As 40.79 mg/kg, Cd 1.3 mg/kg, Cu 214.94 mg/kg, Zn 19.08 mg/kg, Cr 33.29 mg/kg, Sb 0.043 mg/kg. To isolate bacterial strains, soil samples were immediately transferred into the laboratory and serially diluted with sterilized PBS (pH 7.4) solution, then spread on a R2A agar plate containing 0.5 mM CuSO_4_. The plates were then aerobically incubated at 28 °C until visible colonies appeared. Colonies were separately streaked onto a new R2A agar plate containing different concentrations of CuSO_4_ and then incubated at 28 °C for 2~3 days. The streaking process was repeated until pure cultures were obtained. The strain was preliminarily identified by PCR, which targeted the 16S rRNA gene using universal primers 27F (5′-AGAGTTTGATCCTGGCTCAG-3′) and 1492R (5′-TACCTTGTTACGACTT-3′). The purified isolates were cultured in LB medium with a final glycerol concentration of 15%, and then stored at −80 °C.

### 2.3. Determination of Minimum Inhibitory Concentration (MIC) of Heavy Metals and Antibiotics

To determine the MIC of heavy metals and antibiotics, strain C39 was cultured on a solid TMM medium containing sodium gluconate as the sole carbon source and different concentrations of Cu(II), Zn(II), Ni(II), Au(II) or As(II) were added. The antibiotic resistance of *C. necator* strain C39 was also determined by the same method. The MIC was determined in three triplicates as the lowest concentration inhibiting bacterial growth on solid TMM medium containing (1000 mL of ddH_2_O) 2.0 g sodium gluconate, 4.68 g NaCl, 1.49 g KCl, 1.07 g NH_4_Cl, 0.43 g Na_2_SO_4_, 0.2 g MgCl_2_•6H_2_O, 0.03 g CaCl_2_•2H_2_O, 0.23 g Na_2_HPO_4_•12H_2_O, 0.005 g ferric ammonium citrate, 1 mL trace element solution SL7 and 6.06 g Tris, final pH was adjusted to 7.0 using HCl solution [25].

### 2.4. Determination of Growth on Aromatic Compounds

To test the ability for utilization of aromatic compounds by *C. necator* strain C39, several benzene compounds including sodium benzoate, phenol, indole, p-hydroxybenzoic acid, phloroglucinol anhydrous and diphenylamine were used as the sole carbon source. The aromatic compound stock solutions mentioned above were prepared with sterilized double-deionized water and sterilized by filtration using a 0.22 μm micropore membrane. Strain C39 was inoculated into a liquid sterilized carbon-free TMM medium that contains one of the above benzene compounds, and incubated at 28 °C with continuous shaking. The OD_600nm_ of strain C39 was measured at intervals to determine its time-depended growth, so as to verify the ability of strain C39 to degrade aromatic compounds. In addition, carbon-free TMM medium inoculated with strain C39 was used as a control.

### 2.5. Whole Genome Sequencing

Whole genome sequencing of strain C39 was performed on the Illumina and PacBio platforms. In Illumina sequencing, the genomic DNA of strain C39 was extracted and randomly fragmented by sonication. Then the overhangs resulting from fragmentation were converted into blunt ends by using T4 DNA polymerase, Klenow fragment and T4 polynucleotide kinase. After adding an ‘A’ base to the 3′ end of the blunt phosphorylated DNA fragments, adapters were ligated to the ends of the DNA fragments. The desired fragments were purified through gel-electrophoresis, then selectively enriched and amplified by PCR. The index tag was introduced into the adapter at the PCR stage as appropriate. Finally, the qualified library was used for sequencing on a Hiseq 2000 sequencer.

In PacBio sequencing, genomic DNA was first treated with g-TUBE to the appropriate size (>10 kb), then the fragment ends were repaired, and both ends of the DNA fragment were ligated to the connector of the hairpin structure to form a dumbbell structure called SMRTbell. The annealed smrtbell was mixed with the polymerase on the bottom of the Zero-Mode Waveguides (ZWM), which will be used for the final sequencing.

### 2.6. Genome Assembly and Annotation

After filtering the low-quality reads, the remaining clean reads were used for *de novo* assembly using various software including FALCON v. 0.3.0, proovread version 2.12, Celera Assembler version 8.3, SMRT Analysis v2.3.0 and GATK v1.6-13. Default parameter settings were applied in the usage of the software unless otherwise indicated. The final assembly resulted in 2 circular chromosomes and 1 plasmid, with an average coverage of 87×. 

The genome annotation was performed using the NCBI Prokaryotic Genome Annotation Pipeline (PGAP) [26], and the resulting proteome was further annotated with the databases of Carbohydrate-Active enZYmes (CAZy) [27], Cluster of Orthologous Groups (COG) [28], and Kyoto Encyclopedia of Genes and Genomes (KEGG) using the KAAS server [29]. The protein-coding sequences (CDSs) of plasmid were predicted using Glimmer Version: 3.02 [30]. In addition, a rapid annotation of chromosomes and plasmids was also performed using Rapid Annotation using Subsystem Technology (RAST) v2.0 [31]. The features of chromosomes and plasmids were visualized using CGView [32].

### 2.7. Phylogenetic Analysis

Close relative and phylogenetic affiliation of the obtained 16S rRNA sequences were determined by using the BLAST search program at the NCBI website (www.ncbi.nlm.nih.gov) (accessed on 12 July 2022) [33]. The 16S rRNA gene sequences were submitted for comparison and identification to the GenBank databases using the NCBI Blastn algorithm and to the EMBL databases using the Fasta algorithm [34]. To construct a phylogenetic tree, the genome sequence data were uploaded to the Type (Strain) Genome Server (TYGS) (https://tygs.dsmz.de/, accessed on 26 April 2023), a free bioinformatics platform for a whole genome-based taxonomic analysis [35]. Determination of the most closely related type strain genomes was done in two complementary ways by the TYGS: First, the genomes of interest were compared against all type strain genomes available in the TYGS database via the MASH algorithm, a fast approximation of intergenomic relatedness [36], and then, the ten type strains with the smallest MASH distances were chosen per user genome. Second, an additional set of ten closely related type strains was determined via the 16S rDNA gene sequences. These were extracted from the user genomes using RNAmmer [37] and each sequence was subsequently BLASTed [38] against the 16S rDNA gene sequence of each of the currently 14,723 type strains available in the TYGS database. This was used as a proxy to find the best 50 matching type strains (according to the bitscore) for each user genome and to subsequently calculate precise distances using the Genome BLAST Distance Phylogeny approach (GBDP) under the algorithm ‘coverage’ and distance formula d5 [39]. These distances were finally used to determine the 10 closest type strain genomes for the genomes of interest.

In addition, the final assembled sequences were uploaded to the JSpeciesWS Online Service (https://jspecies.ribohost.com/jspeciesws/#analyse, accessed on 26 April 2023) to calculate the average nucleotide identity (ANI) and confirm the closest phylogenetic strain at the genomic level [40], for which the genomes of type strain *Cupriavidus necator* N-1^T^, *C. necator* H16, *C. necator* PHE3-6 and *Cupriavidus lacunae* S23^T^were chosen as references.

### 2.8. RT-qPCR Analysis

Real-time quantitative PCR (RT-qPCR) was employed to test the transcription of *yciI*, *arsI* and the glutathione S-transferase encoding gene (*GST*). The overnight cultures were inoculated into fresh 10 mL of TMM medium at 1% inoculum volume. When the OD_600nm_ reached 0.6, 40 and 200 μM As(III) or 10 μM Rox(III) were added respectively, with no metal addition as control. The cells were harvested after 2 h of induction. Total RNA was extracted using the TRIzol method (TransGen Biotech, TransZol Up Plus RNA Kit), according to the manufacturer’s instructions. The RNA concentrations were quantified using Nanodrop. The synthesis of cDNA from 200 ng of total RNA was performed using the TransScript^®^ One-Step gDNA Removal and cDNA Synthesis SuperMixs (TransGen Biotech, Beijing, China). The resulting cDNA was used as a template for RT-qPCR with the PerfectStart^®^ Green qPCR SuperMix (TransGen Biotech). Primers qB-F/qB-R, qI-F/qI-R, qS-F/qS-R and q39S-F/q39S-R (Appendix A) were used to test the expression of *phnB*, *arsI*, *GST* and 16S rRNA, respectively. The 16S rRNA gene was used as a reference gene to achieve the relative quantification of expression. RT-qPCR was performed using a two-step method (94 °C for 30 s, 94 °C for 5 s, 60 °C for 30 s, 40~45 cycles) following the manufacturer’s recommended protocol. The relative expression was quantified according to the method of 2^−ΔΔCT^ [41].

## 3. Results and Discussion

### 3.1. Minimum Inhibitory Concentration (MIC) Results of Cupriavidus Necator C39

Single colonies of strain C39 were round, convex, opaque, creamy yellow and displayed a moist and smooth surface with flat edges.

We compared the MICs of several heavy metals and antibiotics of *C. necator* strain C39 with three strains of the well characterized heavy metal resistant species *Cupriavidus metallidurans* (*C. metallidurans* strain CH34, *C. metallidurans* strain BS1 and the mega-plasmid free *C. metallidurans* strain AE104) [25,42] (Table 1 and Table 2).

*C. necator* strain C39 was not able to tolerate high concentrations of heavy metal(loid)s as in *C. metallidurans* strain CH34, *C. metallidurans* strain BS1: Cu(II) 2 mM, Zn(II) 2 mM, Ni(II) 0.2 mM, and Au 70 μM in heavy metal salt containing TMM agar plates (Table 1). The only exception was arsenite (As(III)) where a relatively high MIC of 2.5 mM in heavy metal(loid) salt containing TMM agar plates could be determined.

For the antibiotics, *C. necator* strain C39 was able to tolerate more than 256 μg/mL streptomycin and gentamycin, 256 μg/mL kanamycin, >128 μg/mL ampicillin, 32 μg/mL chloramphenicol, 12 μg/mL rifampin and 3 μg/mL tetracycline.

To the best of our knowledge, a characterization of arsenic resistance in *C. necator* has not been reported in previous studies, thus our finding in *C. necator* strain C39 will enhance a better understanding of this species.

### 3.2. Genomic Assembly and Features

The final genome assembly of *C. necator* strain C39 contains 2 circular chromosomes and 1 plasmid, with lengths of 4,077,027 bp, 3,114,252 bp and 1,185,855 bp, respectively. The GC content in chromosomes 1 and 2 was 66.35% and 65.87%, while in the plasmid it was only 62.16% indicating the plasmid or parts of the plasmid were introduced more recently in evolutionary time. It is a common feature that all sequenced *Cupriavidus* species have multi-replicon genomes, often including large plasmids [18,43,44,45]. A total number of 7374 protein coding CDSs, 15 rRNAs, 65 tRNAs and 42 sRNAs were identified in the whole genome of strain C39 (Figure 1). Among the total number of 7960 predicted genes, 6249 genes accounting for 78.5% are annotated based on the COG database (Figure 2a), 5050 genes accounting for 63.44% are annotated based on the GO database (Figure 2b), and 4416 genes accounting for 55.47% are annotated based on KEGG database (Figure 2c), respectively. In addition, a total of 77 genes are annotated in the CAZy database, which indicated that *C. necator* strain C39 is likely to utilize various organic carbon sources for respiration, and 36 genes are annotated in the ARDB database, suggesting *C. necator* strain C39 is able to tolerate a number of antibiotics. 

### 3.3. Phylogenetic Characterization

The phylogenetic tree based on the 16S rRNA gene revealed that strain C39 is a member of *Cupriavidus* with its closest relatives, which include the strain *C. necator* UYPR2.512, *C. necator* NH9 and *C. lacunae* S23^T^ (Figure 3). The whole genome-based taxonomic analysis using the Genome BLAST Distance Phylogeny (GBDP) provided by the Leibniz Institute dSMZ (https://tygs.dsmz.de/, accessed on 26 April 2023) showed that *C. necator* C39 was closest to *C. necator* N-1 and *C. necator* KK10 (Figure 3). However, *C. necator* strain C39 showed the highest average nucleotide identity (ANI) value of 94.52% with *C. necator* N-1^T^, and the ANI value with *C. necator* H16 and *C. necator* PHE3-6 was 92.70% and 92.44%, respectively. ANI value between strain C39 and *C. lacunae* S23^T^ was only 89.82%. Researchers proposed that standard ANI cut-off values of 90% should be applied to *Cupriavidus* strains [18], according to this cut-off, it is clear that strain C39 belongs to *Cupriavidus necator*, and ANI analysis confirmed the validity of the species *C. lacunae*, since *C. lacunae* S23^T^ only showed an ANI value of 88.86% with *C. necator* N-1^T^ [46]. 

### 3.4. Degradation of Aromatic Compounds

*C. necator* strain C39 was able to grow on TMM medium containing benzoate, phenol, indole, p-hydroxybenzoic acid or phloroglucinol anhydrous as the sole carbon source, while growth did not occur on TMM medium containing diphenylamine nor on carbon-free TMM medium (Figure 4). These results indicated that *C. necator* strain C39 was able to degrade benzene compounds including benzoate, phenol, indole, p-hydroxybenzoic acid and phloroglucinol anhydrous.

### 3.5. Functional Annotations

According to the NCBI pipeline annotation results, different metal homeostasis related genes could be identified based on the annotation of these gene products. Several copper tolerance related genes, i.e., the copper resistance protein CopA and CopB (Table 3), were identified in the genome of strain C39. In addition, the cobalt-zinc-cadmium resistance protein CzcD, a probable Co/Zn/Cd efflux system membrane fusion protein CzsB and a Zn(II) and Co(II) transmembrane diffusion facilitator CzrB were present in multiple copies (Table 3), those genes may contribute to the high resistance of *C. necator* strain C39 to Cu(II) and Zn(II), which is very different from another heavy metal resistant bacterium *Cupriavidus campinensis* S14E4C [47]. Although *C. necator* strains are often resistant to high levels of copper, resistance to high concentrations of As(III) has rarely been observed in other strains. Arsenic-resistance (*ars*) operons or clusters are widely distributed in *Burkholderiales* genomes in diverse combinations [48]. The *ars* cluster *arsR-arsICBR-yciI* was identified on the genome of *C. necator* strain C39, which has a similar organization as in *C. necator* N-1^T^, *C. necator* KK10, *C. necator* H850 and *C. necator* H16 (Figure 5). It is noteworthy that a glutathione S-transferase (*GST*) was recruited in *C. necator* strain C39 and another two *Cupriavidus* strains (*C. necator* B9 and *Cupriavidus* sp. SK4), which may enhance the resistance to arsenate for bacterium [49]. RT-qPCR results indicated that *arsI*, which is a representative gene in the *ars* operon of *C. necator* strain C39 and essential for arsenate reduction [50], was upregulated in the presence of both concentrations of As(III) and roxarsone(III) (also Rox(III)). Transcription of *yciI* and *GST*, which were adjacent upstream and downstream of the *ars* operon, were also upregulated in the presence of 40 μM or 200 μM As(III) and 10 μM of Rox(III) (Figure 6). The function of the encoded gene product YciI is unclear but related genes have been involved in lyase activity [51,52]. Possibly, *yciI* encoded on many *ars* clusters could function as a C-As lyase detoxifying organic As-compounds. In addition, there is a scattered gene encoded for an arsenite efflux pump ArsB together with a gene of an arsenic transporter located in other loci. Considering arsenic often appears as an associated mineral in gold mines [23], it was not surprising that *C. necator* strain C39 was able to tolerate high concentrations of As(III).

Among the 36 annotated antibiotic resistance genes (ARGs), a majority (21 out of 36) of them belong to multidrug resistance efflux pump (Appendix A), which is mainly responsible for the resistance of aminoglycoside, tigecycline, fluoroquinolone, beta-lactam, tetracycline, glycylcycline, macrolide, acriflavin and chloramphenicol. Considering the high resistance to streptomycin, gentamycin, kanamycin and ampicillin for strain C39, it is possible that resistance to antibiotics for this bacterium may attribute to those multidrug resistance efflux pumps. However, it has been reported that heavy metal pollution increases metal resistance and reduces antibiotic sensitivity due to co-regulation of genes [53], it is also possible that the isolation of strain C39 from heavy metal enriched environment promoted the heavy metal and antibiotic resistance as well. 

In addition, the metabolic pathway for aromatic compound degradation was reconstructed using the KEGG database. Pathway analysis suggested that strain C39 should be able to completely degrade benzoate, benzamide, catechol, 3- or 4-fluorobenzoate, 3- or 4-hydroxybenzoate, 3,4-dihydroxybenzoate and phenol (Table 4). Compared to *C. necator* NH9, strain C39 contains the complete set of genes for phenol-degrading [18]. Key genes that are responsible for hydrogen utilization and nitrogen fixation, i.e., genes encoding hydrogenase and nitrogenase, were not detected on the genome of *C. necator* strain C39, indicating that this strain was not able to grow chemolithoautotrophically and did not have the ability to fix nitrogen. In addition, key genes for poly(3-hydroxybutyrate) [P(3HB)] synthesis, which is a representative member of biodegradable polyesters, were not identified on the genome of *C. necator* strain C39 [5].

The function-based comparison of RAST was used to compare the similar and different genes between *C. necator* strain C39 and *C. metallidurans* strain CH34 in different categories. Under the category “Metabolism of aromatic compounds”, strain C39 has 10 unique roles which belong to benzoate degradation, biphenyl degradation, aromatic amine catabolism, gentisate degradation and other subsystems respectively. This, at the genetic level, explains why strain C39 had a strong ability to degrade benzoate in previous experiments. Under the category “virulence, disease and defense”, the common genes of the two can be subdivided into 39 roles, and the unique genes of strain C39 only have 3 roles, while strain CH34 has 12 roles. Strain CH34 has more genes related to copper homeostasis and cobalt, zinc and cadmium resistance than strain C39, which can also be used to explain why its MIC results are higher than strain C39. At the same time, we speculate that although strain C39 also has many RNDs, these RNDs may not all play the role of drug resistance and cannot be used for heavy metal efflux, so its MIC results are lower than strain CH34.

## 4. Conclusions

Our study displays the presence of multiple heavy metals and antibiotic-resistance determinants *Cupriavidus necator* C39, isolated from a gold-copper mine. Interestingly, strain C39 has shown significant resistance to some metals and antibiotics and also the potential to degrade aromatic compounds such as benzoate, phenol, indole, p-hydroxybenzoic acid or phloroglucinol anhydrous. The whole genome analysis of strain C39 revealed multiple genes predicted to encode functions responsible for metal and antibiotic resistance and the degradation pathway of aromatic compounds. Among these genes, the arsenic-resistance (*ars*) cluster *GST-arsR-arsICBR-yciI* and a scattered gene encoding ArsB is predicted to confer arsenic resistance; genes encoding multidrug resistance efflux pump are predicted to confer high antibiotic resistance to strain C39. Moreover, the presence of a number of genes predicted to encode key enzymes in the degradation pathway of benzene compounds indicate the verified potential of strain C39 to degrade these compounds. 

## Figures and Tables

**Figure 1 microorganisms-11-01518-f001:**
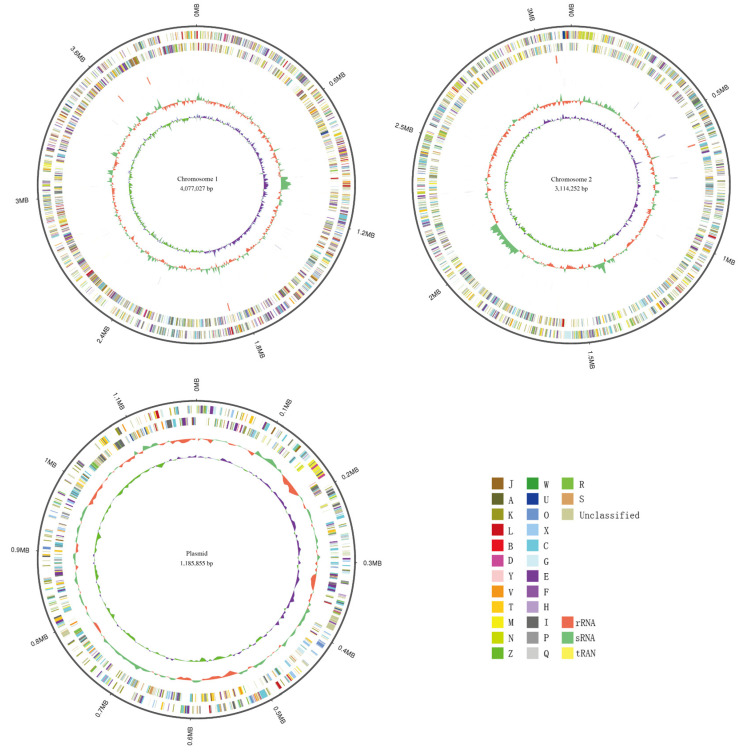
Characteristics of the chromosomes and plasmid of *C. necator* C39. From outside to inside: genome size, forward strand, colored according to cluster of orthologous groups (COG) classification, reverse strand, colored according to COG classification, GC skew, GC content.

**Figure 2 microorganisms-11-01518-f002:**
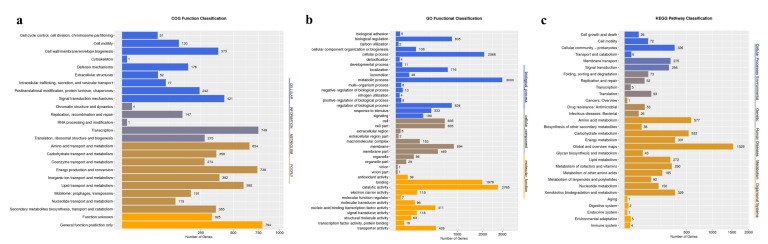
Statistics of annotated genes based on COG database (**a**), GO database (**b**) and KEGG database (**c**).

**Figure 3 microorganisms-11-01518-f003:**
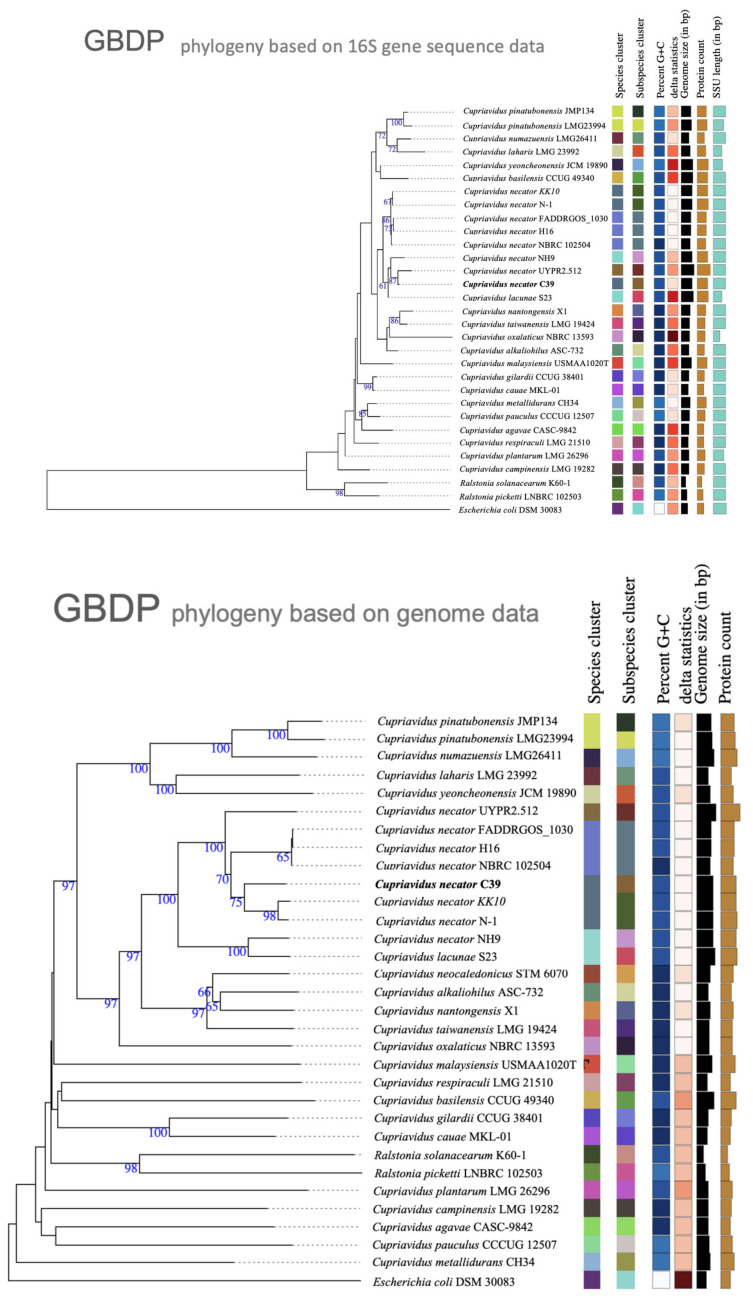
A molecular phylogenetic tree based on the MASH algorithm and 16S rRNA gene sequences highlighting the position of *C. necator* C39 relative to other type and non-type strains of the genus *Cupriavidus* and *Ralstonia*, as outgroup the genus *Escherichia*. The evolutionary history was inferred by MASH and 16S rDNA-based tree with the Type (Strain) Genome Server (TYGS) uses the Genome BLAST Distance Phylogeny (GBDP) provided by Leibniz Institute DSMZ (https://tygs.dsmz.de/, accessed on 26 April 2023), for a whole genome-based taxonomic analysis. Color code; Spieces & subspieces cluster—Matching colors describe same species, Percent G+C—The darker the shade, the higher the G+C content, delta statistics—provide guidance regarding the suitability of specific query genome sequences and the reliability of the phylogenetic outcome, Genome size, protein counts and SSU—The bar length reflects the relative size between the use species and strains.

**Figure 4 microorganisms-11-01518-f004:**
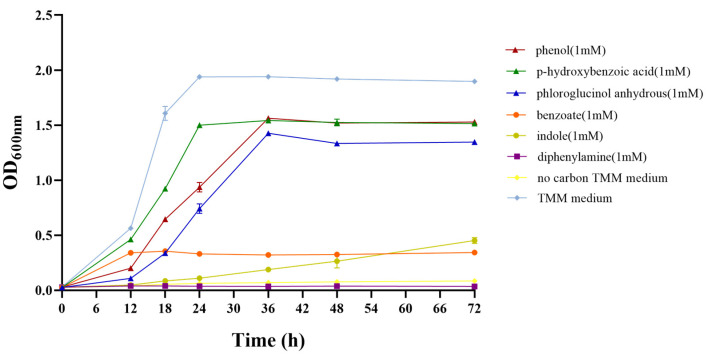
Growth of strain C39 on aromatic compounds. Error bars represent the estimated standard deviations for triplicate samples. *C. necator* strain C39 was able to grow on TMM medium that contains benzoate, phenol, indole, p-hydroxybenzoic acid or phloroglucinol anhydrous as the sole carbon source, while growth neither occurred on TMM medium containing diphenylamine nor on carbon-free TMM medium.

**Figure 5 microorganisms-11-01518-f005:**
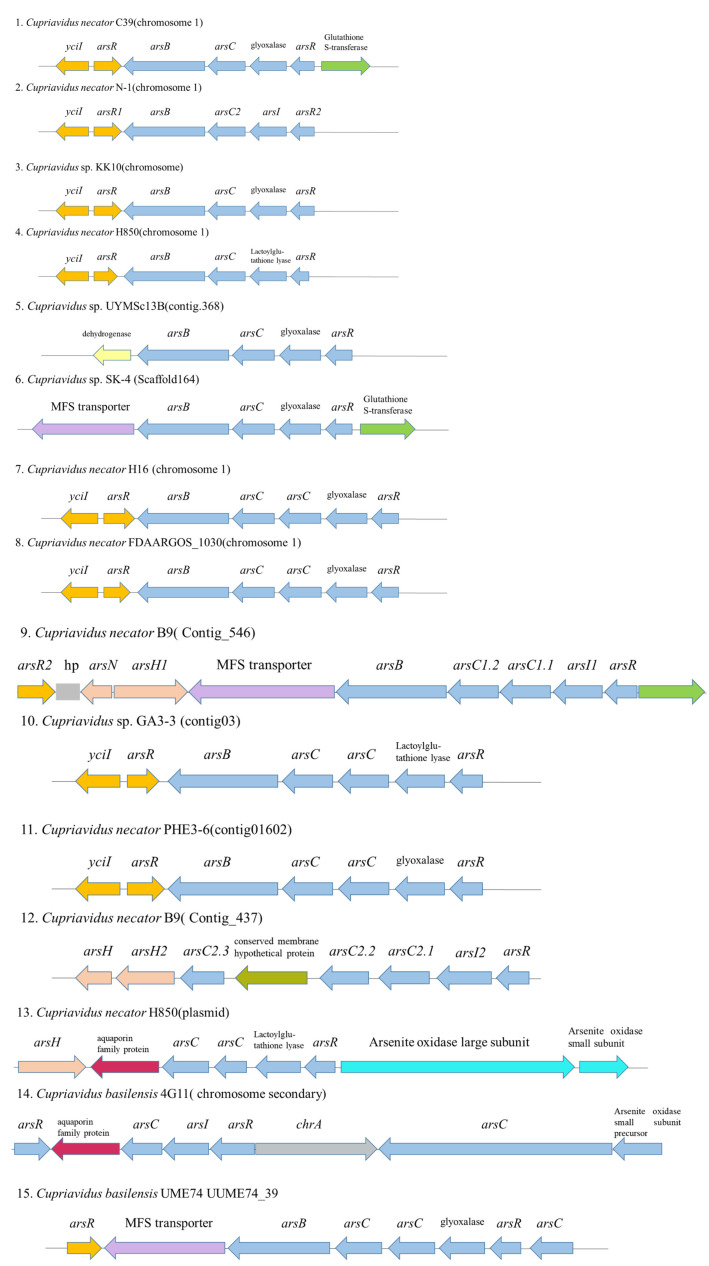
Organizations of the arsenate-resistance operon (*ars*) cluster in *C. necator* C39 and reference genomes. Gene symbols: *arsR*, metalloregulator ArsR/SmtB family transcription factor; *arsC*, arsenate reductase; *arsB*, ACR3 family arsenite efflux transporter; *arsH*, arsenical resistance protein; *yciI,* potential C-As lyase; *arsI,* glyoxylase or lactoylglutathinone lyase; *GST*, glutathione-S-transferase, MFS major facilitator superfamily.

**Figure 6 microorganisms-11-01518-f006:**
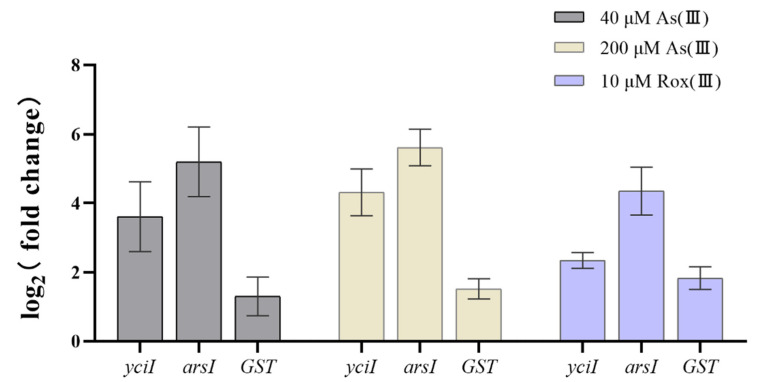
The effect of As(III) and Rox(III) to the transcription of *yciI*, *arsI* and *GST*. For RT-qPCR, error bars correspond to the standard deviations of the means from three biological replicates. Gene expression was normalized to the 16S rRNA gene. The results are presented as the mean gene expression normalized to mRNA levels in As(III)/Rox(III)-free TMM. Data symbols shown in all panels are the same.

**Table 1 microorganisms-11-01518-t001:** Minimum Inhibitory Concentration (MIC) results of *Cupriavidus* performed in triplicate.

Heavy Metals	C39	CH34	BS1	AE104
As(III)	2.5 mM	2.5 mM	3 mM	2.5 mM
Au(III)	70 μM	100 μM	110 μM	80 μM
Cu(II)	2 mM	4.5 mM	3 mM	2.5 mM
Zn(II)	2 mM	12 mM	12 mM	0.5 mM
Ni(II)	0.2 mM	6 mM	9 mM	0.5 mM

**Table 2 microorganisms-11-01518-t002:** Minimum Inhibitory Concentration (MIC) of selected antibiotics of *Cupriavidus*.

Antibiotics	C39	CH34	BS1	AE104
kanamycin	256 μg/mL	>256 μg/mL	>256 μg/mL	>256 μg/mL
streptomycin	>256 μg/mL	>256 μg/mL	>256 μg/mL	>256 μg/mL
gentamycin	>256 μg/mL	>256 μg/mL	>256 μg/mL	>256 μg/mL
ampicillin	>128 μg/mL	>128 μg/mL	>128 μg/mL	>128 μg/mL
chloramphenicol	32 μg/mL	>256 μg/mL	>256 μg/mL	>256 μg/mL
rifampin	12 μg/mL	64 μg/mL	28 μg/mL	64 μg/mL
tetracycline	3 μg/mL	4 μg/mL	5 μg/mL	4 μg/mL

**Table 3 microorganisms-11-01518-t003:** Genes/gene clusters of heavy metal(loid) resistance in *C. necator* C39.

Genes	Functions	Metal(loid)	Locus Tag
*cusA*	efflux RND transporter permease subunit	Cu	JJQ59_11025
*cusB*	efflux RND transporter periplasmic adaptor subunit	Cu	JJQ59_11020
*cusC*	TolC family protein	Cu	JJQ59_11015
*PCuC*	copper chaperone PCu(A)C	Cu	JJQ59_01040
*copQ* _1_	copper resistance protein1	Cu	JJQ59_37555
*copQ* _2_	copper resistance protein	Cu	JJQ59_23855
*copS*	heavy metal sensor histidine kinase	Cu	JJQ59_26675
*copS*	heavy metal sensor histidine kinase	Cu	JJQ59_30460
*copR*	heavy metal response regulator transcription factor	Cu	JJQ59_30455
*copA*	copper resistance system multicopper oxidase	Cu	JJQ59_30450
*copB*	copper resistance protein B	Cu	JJQ59_30445
*copC*	copper homeostasis periplasmic binding protein	Cu	JJQ59_30440
*copD*	copper homeostasis membrane protein	Cu	JJQ59_30435
*ctpA*	copper translocating P-type ATPase	Cu	JJQ59_11645
*ctpA*	copper translocating P-type ATPase	Cu	JJQ59_10950
*cueR*	Cu(I)-responsive transcriptional regulator	Cu	JJQ59_18470
*ctpA*	copper-translocating P-type ATPase	Cu	JJQ59_18475
*copZ*	heavy-metal-associated domain-containing protein	Cu	JJQ59_18480
*atpX*	heavy metal translocating P-type ATPase		JJQ59_35770
*czcB*	efflux RND transporter periplasmic adaptor subunit	Co/Zn/Cd	JJQ59_31505
*czcC*	TolC family protein	Co/Zn/Cd	JJQ59_31510
*czcI*	cobalt-zinc-cadmium resistance protein	Co/Zn/Cd	JJQ59_31515
*czcA*	efflux RND transporter permease subunit	Co/Zn/Cd	JJQ59_27715
*czcB*	efflux RND transporter periplasmic adaptor subunit	Co/Zn/Cd	JJQ59_27720
*czcC*	TolC family protein	Co/Zn/Cd	JJQ59_27725
*czcI*	cobalt-zinc-cadmium resistance protein	Co/Zn/Cd	JJQ59_27730
*czcC*	TolC family protein	Co/Zn/Cd	JJQ59_20825
*czcB*	efflux RND transporter periplasmic adaptor subunit	Co/Zn/Cd	JJQ59_20830
*czcA*	efflux RND transporter permease subunit	Co/Zn/Cd	JJQ59_20835
*czcC*	TolC family protein	Co/Zn/Cd	JJQ59_35985
*czcA*	efflux RND transporter permease subunit	Co/Zn/Cd	JJQ59_35990
*czcB*	efflux RND transporter periplasmic adaptor subunit	Co/Zn/Cd	JJQ59_35995
*qseC*	sensor histidine kinase N-terminal domain-containing protein		JJQ59_36000
*qseB*	winged helix-turn-helix domain-containing protein		JJQ59_36005
	efflux RND transporter periplasmic adaptor subunit		JJQ59_24830
	efflux RND transporter permease subunit		JJQ59_24835
	efflux RND transporter periplasmic adaptor subunit		JJQ59_31600
*dmeF*	CDF family Co(II)/Ni(II) efflux transporter DmeF		JJQ59_01450
	Cobalt-zinc-cadmium resistance protein	Co/Zn/Cd	JJQ59_17920
*zntA*	heavy metal translocating P-type ATPase	Zn/Cd	JJQ59_26665
*zntR*	Cd(II)/Pb(II)-responsive transcriptional regulator	Zn/Cd	JJQ59_18160
*mgtA*	magnesium-translocating P-type ATPase	Mg	JJQ59_37705
*corA*	magnesium/cobalt transporter(uptake system)	Mg/Co	JJQ59_16020
*dmeF*	CDF family Co(II)/Ni(II) efflux transporter DmeF	Ni/Co	JJQ59_01450
*arsC*	arsenate reductase (glutaredoxin)	As	JJQ59_18050
	arsenic transporter	As	JJQ59_29185
*arsB*	arsenite efflux pump	As	JJQ59_29190
*arsB*	ACR3 family arsenite efflux transporter	As	JJQ59_10970
*arsC*	arsenate reductase	As	JJQ59_10975
*arsI*	glyoxalase/bleomycin resistance/dioxygenase family protein	As	JJQ59_10980
*arsR*	metalloregulator ArsR/SmtB family	As	JJQ59_10985
*arsR* _1_	metalloregulator ArsR/SmtB family	As	JJQ59_10965
*phnB*	YciI family protein; putative C-As lyase	As	JJQ59_10960
*GST*	glutathione S-transferase family protein	As	JJQ59_10990

**Table 4 microorganisms-11-01518-t004:** Putative genes of *C. necator* C39 involved in degradation of aromatic and relative compounds.

Compound Name	Gene Symbol	K Number	EC Number	Definition	Annotated Genes
benzoate	*benA*	K05549	EC:1.14.12.10	benzoate 1,2-dioxygenase alpha subunit	JJQ59_09715 (Chr 1)
	*benB*	K05550	EC:1.14.12.10	benzoate 1,2-dioxygenase beta subunit	JJQ59_09710 (Chr 1)
	*benD*	K05783	EC:1.3.1.25	1,6-dihydroxycyclohexa-2,4-diene-1-carboxylate dehydrogenase	JJQ59_09700 (Chr 1)
3-hydroxybenzoate	*nagX*	K22270	EC:1.14.13.24	3-hydroxybenzoate 6-monooxygenase	JJQ59_22370 (Chr 2)
	*nagI*	K00450	EC:1.13.11.4	gentisate 1,2-dioxygenase	JJQ59_22355 (Chr 2)
	*nagL/maiA*	K01801	EC:5.2.1.2	maleylpyruvate isomerase/maleylacetoacetate isomerase	JJQ59_22365 (Chr 2)
	*nagK/NA*	K16165	EC:3.7.1.20	fumarylpyruvate hydrolase/fumarylacetoacetate hydrolase	JJQ59_22365 (Chr 2)
4-hydroxybenzoate	*pobA*	K00481	EC:1.14.13.2	4-hydroxybenzoate 3-monooxygenase	JJQ59_30915 (Chr 2)
	*pcaG*	K00448	EC:1.13.11.3	protocatechuate 3,4-dioxygenase, alpha subunit	JJQ59_30935 (Chr 2)
	*pcaH*	K00449	EC:1.13.11.3	protocatechuate 3,4-dioxygenase, beta subunit	JJQ59_30940 (Chr 2)
	*pcaB*	K01857	EC:5.5.1.2	3-carboxy-cis,cis-muconate cycloisomerase	JJQ59_30930 (Chr 2)
	*pcaC*	K01607	EC:4.1.1.44	carboxymuconolactone decarboxylase family	JJQ59_25245 (Chr 2)
	*pcaD*	K14727	EC:3.1.1.24	3-oxoadipate enol-lactonase	JJQ59_09740 (Chr 2), JJQ59_26420 (Chr 2)
catechol	*catA*	K03381	EC:1.13.11.1	catechol 1,2-dioxygenase	JJQ59_09720 (Chr 1)
	*catB*	K01856	EC:5.5.1.1	muconate cycloisomerase	JJQ59_20615 (Chr 2)
	*catC*	K03464	EC:5.3.3.4	muconolactone delta-isomerase	JJQ59_09735 (Chr 1)
	*pcaD*	K01055	EC:3.1.1.24	3-oxoadipate enol-lactonase	JJQ59_30925 (Chr 2)
phenol	*dmpK*	K16249	EC:1.14.13.244	phenol hydroxylase P0 protein	JJQ59_20635 (Chr 2)
	*dmpL*	K16243	EC:1.14.13.244	phenol hydroxylase P1 protein	JJQ59_20640 (Chr 2)
	*dmpM*	K16244	EC:1.14.13.244	phenol hydroxylase P2 protein	JJQ59_20645 (Chr 2)
	*dmpN*	K16242	EC:1.14.13.244	phenol hydroxylase P3 protein	JJQ59_20650 (Chr 2)
	*dmpQ*	K16245	EC:1.14.13.244	phenol hydroxylase P4 protein	JJQ59_20655 (Chr 2)
	*dmpP*	K16246	EC:1.14.13.244	phenol hydroxylase P5 protein	JJQ59_20660 (Chr 2)
benzonitrile	NA	K01501	EC: 3.5.5.1	nitrilase	JJQ59_09680 (Chr 1)
benzamide	*amiE*	K01426	EC:3.5.1.4	amidase	JJQ59_09300 (Chr 1)

## Data Availability

The chromosome and plasmid sequences of *C. necator* C39 can be accessed under the GenBank accession number CP068434.1, CP068435.1 and CP068436.1 respectively. The data sets of PacBio and Illumina reads are available in the NCBI SRA database via the BioProject accession number PRJNA690866.

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
