# Peer review of "Whole Genome Sequence Analysis of Cupriavidus necator C39, a Multiple Heavy Metal(loid) and Antibiotic Resistant Bacterium Isolated from a Gold/Copper Mine"

_microorganisms, 2023, doi:10.3390/microorganisms11061518_

Round 1

Reviewer 1 Report

Lines 16-17 - Sentence is looks strange, predicate is missed.

Line 22 - "... sole carbon sources" 

Line 24 - space is missed

Line 59 - gold mines

In my opinion introduction section should be bit expanded with brief data on physiology and genetics of the published Cupriavidus necator strains data and biodiversity of heavy-metal resistant bacteria and their biotechnological applications.

Lines 75-76 - Heavy metal content in the soil studied. By which method this data were obtained? Should be mentioned.

Lines 78-79 - incubation period in culturing should be mentioned in days, approximately at least.

Line 84 - Glycerol stock concentration should be added

Lines 140, 144, 160 - url-link should be formatted as reference

Lines 178-180 - Sentence should be revised for grammar. 

Section 3.1 have no discussion. Comparesment with the other known heavy-metal tolerant bacteria and previously described resistant Cupriavidus strains would be nice.

Figures 1 - 3 & 5 should be formatted to be well-included into the manuscript body. Figures titles should be formatted properly.

Short conclusion section should be added for readers to summarise the most interesting properties of the strain sequenced found.

English looks fine. Somewhere sentences are incomplete, need to be revised. I was tried to mark them in review report.

Author Response

Point-by-point response to the comments from the reviewers

Reviewer #1:

  1. Lines 16-17 - Sentence is looks strange, predicate is missed.

Response: Thanks for the comment. Changes have been made in the new revised text in the line 17-19.

  1. Line 22 - "... sole carbon sources"

Response: Thanks for the comment. We have considered the change in line 22.

  1. Line 24 - space is missed

Response: Thanks for the comment. The change has been made in the requested line 24.

  1. Line 59 - gold mines

Response: Thanks for the comment. Change has been made in the requested line 68.

  1. In my opinion introduction section should be bit expanded with brief data on physiology and genetics of the published Cupriavidus necator strains data and biodiversity of heavy-metal resistant bacteria and their biotechnological applications.

Response: Thanks for the comment. The physiology and genetics of the published Cupriavidus necator strains have been summarized in the first paragraph, here we added the general description of this species, and we also added a brief introduction on the biodiversity of heavy-metal resistant bacteria and their biotechnological applications. The new inputs have been made in lines 42-43 and 56-63.

  1. Lines 75-76 - Heavy metal content in the soil studied. By which method this data were obtained? Should be mentioned.

Response: Thanks for the comment. We have added the relevant information as requested in Lines 90 to 93.

  1. Lines 78-79 - incubation period in culturing should be mentioned in days, approximately at least.

Response: Thanks for the comment. The relevant information has been added as requested in line 99-100.

  1. Line 84 - Glycerol stock concentration should be added

Response: Thanks for the comment. The relevant information has been added as requested in line 103-105.

  1. Lines 140, 144, 160 - url-link should be formatted as reference

Response: Thanks for the comment. We have made the relevant modification as requested in line 162-165.

  1. Lines 178-180 - Sentence should be revised for grammar.

Response: Thanks for the comment. We have made changes in line 198-200.

  1. Section 3.1 have no discussion. Comparesment with the other known heavy-metal tolerant bacteria and previously described resistant Cupriavidus strains would be nice.

Response: Thanks for the comments. Here we only compared the characteristics of heavy metals and antibiotics resistance between C. necator strain C39 and C. metallidurans strains. To the best of our knowledge, a characterization of arsenic resistance in C. necator has not been reported in previous studies, thus our finding in C. necator strain C39 will enhance a better understanding about this species. However, we have highlighted the information relevant to our finding in lines 221-223.

  1. Figures 1 - 3 & 5 should be formatted to be well-included into the manuscript body. Figures titles should be formatted properly.

Response: Thanks for the comment. We have revised these formatting issues in the revised manuscript.

  1. Short conclusion section should be added for readers to summarize the most interesting properties of the strain sequenced found.

Response: Thanks for the comment. A conclusion section has been added to the manuscript from line 369 to 382.  

  1. English looks fine. Somewhere sentences are incomplete, need to be revised. I was tried to mark them in review report.

Response: Thanks for the comment. We have made the required changes.

Reviewer 2 Report

The study isolated a bacterium C39 with resistance to a variety of heavy metals and antibiotics. Moreover, strain C39 was able to grow on TMM medium containing with aromatic compounds like benzoate, phenol, indole, P-hydroxybenzoic acid or phloroglucinol anhydrous as the sole carbon source. The entire genome of the bacterium was sequenced and analyzed. Genes associated with organo-arsenic resistance were identified. In addition, genes related to the benzene compounds degradation pathway were also identified, suggesting that this bacterium could be used for the degradation of benzene compounds. These findings are interesting. However, I suggest that there are some details that need further clarification.

1. The purpose of this study needs to be further clarified and clarified to reflect the novelty and uniqueness of this study.

2.The bacteria were isolated from soils contaminated with a variety of heavy metals, especially Cu, which was particularly high. The copper-containing medium was also used in subsequent isolation experiments. My guess is that more than one type of bacteria was purified and isolated, but the author did not give a detailed account of the results of bacterial isolation. Is this method of isolating bacteria consistent with the goal of your original experiment?

3.line 97, This experiment can only show the tolerance of the bacteria to benzene compounds, but can not test its ability to degrade aromatic compounds.

4. line 187, 189. Table 1 and 2 do not show the difference in characteristics between this strain and other bacteria. Does this bacterium have new functional characteristics?

5. Some key genes related to the phenol degradation pathway were found in the genome sequence analysis, it can be considered to verify the relevant enzyme activity through biochemical experiments, or to prove that the bacteria have this ability.

Author Response

Reviewer #2:

The study isolated a bacterium C39 with resistance to a variety of heavy metals and antibiotics. Moreover, strain C39 was able to grow on TMM medium containing with aromatic compounds like benzoate, phenol, indole, P-hydroxybenzoic acid or phloroglucinol anhydrous as the sole carbon source. The entire genome of the bacterium was sequenced and analyzed. Genes associated with organo-arsenic resistance were identified. In addition, genes related to the benzene compounds degradation pathway were also identified, suggesting that this bacterium could be used for the degradation of benzene compounds. These findings are interesting. However, I suggest that there are some details that need further clarification.

Response: We appreciate the reviewer’s comments and have tried our best to address the reviewer’s concerns as following:

  1. The purpose of this study needs to be further clarified and clarified to reflect the novelty and uniqueness of this study.

Response: Thanks for the comment. We have considered this comment and we made the relevant modification in line 68-74.

  1. The bacteria were isolated from soils contaminated with a variety of heavy metals, especially Cu, which was particularly high. The copper-containing medium was also used in subsequent isolation experiments. My guess is that more than one type of bacteria was purified and isolated, but the author did not give a detailed account of the results of bacterial isolation. Is this method of isolating bacteria consistent with the goal of your original experiment?
  2. line 97, This experiment can only show the tolerance of the bacteria to benzene compounds, but can not test its ability to degrade aromatic compounds.

Response: Thanks for the comment. According to the result of growth experiments as shown in Figure 4, when compared to carbon-free TMM medium, growth only occurred in the presence of aromatic compounds as the sole carbon source, thus indicating that the added aromatic compounds were utilized by the bacterium to support growth, and in this sense, we propose that the bacterium is able to degrade those aromatic compounds instead of merely tolerate to them because they need a carbon source. In future experiments we aim to further characterize the metabolism of aromatic compounds and thereby detect the concentration decrease of the aromatic compounds. To be more precise, we have revised the sentence “which now assigned in line 117”.

  1. line 187, 189. Table 1 and 2 do not show the difference in characteristics between this strain and other bacteria. Does this bacterium have new functional characteristics?

Response: Thanks for the comment. Although there appears to not be much difference when looking at heavy metal and antibiotic resistances between strain C39 and other bacteria, we would like to point out that strain C39 belong to C. necator, while the other strains belong to C. metallidurans, which is one of the most heavy metal resistant bacteria that have been studied. Moreover, and the best of our knowledge, the characteristics of arsenic resistance in C. necator has not been studied. We have added discussion in the revised manuscript.

  1. Some key genes related to the phenol degradation pathway were found in the genome sequence analysis, it can be considered to verify the relevant enzyme activity through biochemical experiments, or to prove that the bacteria have this ability.

Response: Thanks for the comment and this suggestion. The current study did not include these experiments but we are planning to perform the biochemical characterization of key enzymes involved in the degradation pathways of aromatic compounds in future experiments.

Reviewer 3 Report

The authors presented a manuscript to analyze the whole genome, multiple heavy metal (loid) and antibiotic resistant of a Cupriavidus necator strain isolated from a gold/copper mine. They found this strain can tolerate to intermediate concentrations of heavy metal and multiple antibiotics, and it can use aromatic compounds such as benzoate. The key genes encoding putative arsenite efflux pump, multidrug resistant efflux pump and degradation pathways of benzene compounds were identified.  It an interesting works. My comments are in following.

1)      Line 16: Cupriavidus necator C39 (C. necator C39)

2)      Line 68-70: Rifampicin is extremely insoluble in water.

3)      Line 78-80: Please describe the morphology of colonies or provide a picture.

4)      Line 81-84: how to identify the strain, sequencing or ?

5)      Setion 2.4: Did the authors monitor the concentration of those benzene?

6)      Line 194-195: I think the MIC of arsenite was not higher than that of other strains.

7)      Section 3.4: How about other C. necator strains? Can they use those aromatic compounds? I really hope to see the data on the concentration of aromatic compounds in the culture medium decreasing over time.

8)      Fig.6: 2000uM As(III) or 200uM As(III)? Please check it.

9)      Please add a conclusion.

Author Response

Reviewer #3:

The authors presented a manuscript to analyze the whole genome, multiple heavy metal (loid) and antibiotic resistant of a Cupriavidus necator strain isolated from a gold/copper mine. They found this strain can tolerate to intermediate concentrations of heavy metal and multiple antibiotics, and it can use aromatic compounds such as benzoate. The key genes encoding putative arsenite efflux pump, multidrug resistant efflux pump and degradation pathways of benzene compounds were identified.  It an interesting works. My comments are in following.

Response: Thanks for the comment. Our answers to the different comments and suggestions are as following:

  1. Line 16: Cupriavidus necator C39 (C. necator C39)

Response: Thanks for the comment. We have added the abbreviation of the species next to the full name as suggested.

  1. Line 68-70: Rifampicin is extremely insoluble in water.

Response: Thanks for the comment. We have corrected the mistake and we added the relevant information in line 84-87.

  1. Line 78-80: Please describe the morphology of colonies or provide a picture.

Response: Thanks for the comment. We have added a description of C. necator C39 colonies in line 203-204.

  1. Line 81-84: how to identify the strain, sequencing or?

Response: Thanks for the comment. We identified the strain by sequencing the 16S rRNA gene of the strain and mapping the phylogenetic tree based on this gene, the results showed that our isolate C39 belongs to the genus Cupriavidus. The whole genome-based taxonomic analysis using the Genome BLAST Distance Phylogeny showed that strain C39 was most closely related to C. necator N-1 and C. necator KK10. Specifically, C39 showed the highest ANI value of 94.52% with C. necator N-1T and according to the standard ANI cut-off values of 90%, isolate C39 was Cupriavidus necator. We have added made changes in line 101-103.

  1. Setion 2.4: Did the authors monitor the concentration of those benzene?

Response: Thanks for the comment. We did not perform this experiment in this current study. However, we are adding this as part for our subsequent studies.

  1. Line 194-195: I think the MIC of arsenite was not higher than that of other strains.

Response: Thanks for the comment. In this section, we compared the MICs of several heavy metals of strain C39 with three strains of the well characterized heavy metal resistant genus Cupriavidus metallidurans, one of the most heavy metal resistant bacterial species. What we want to convey is that with the exception of arsenite, the MICs of other heavy metals of strain C39 are significantly lower than that of other strains of C. metallidurans.

  1. Section 3.4: How about other C. necator strains? Can they use those aromatic compounds? I really hope to see the data on the concentration of aromatic compounds in the culture medium decreasing over time.

Response: Thanks for the comment. Some strains of the genus Cupriavidus necator have also been shown to degrade aromatic compounds. We are performing analytical work on the quantity of the consumed benzene and possible degradation or transformation products for our future study.

  1. Fig.6: 2000uM As(III) or 200uM As(III)? Please check it.

Response: Thanks for the comment. It is indeed a typo and we have done the correction.

  1. Please add a conclusion.

Response: Thanks for the comment. A concluding section has been added from line 369-382.

Round 2

Reviewer 3 Report

The authors have addresses my comments.

Author Response

Letter to Editor

RE: Revised Manuscript microorganisms-2391139-2

Dear Editor,

Thank you for your comments and for giving us the opportunity to resubmit our revised manuscript entitled “Whole genome sequence analysis of Cupriavidus necator C39, a multiple heavy metal(loid) and antibiotic resistant bacterium isolated from a gold/copper mine”. We also thank the reviewers for their constructive comments and suggestions concerning our manuscript. We have considered and responded to all comments and suggestions raised by the reviewers, and have revised them. Our point-by-point responses to the reviewer and editor comments are attached below.

We greatly appreciate your time and effort.

Best regards,

Christopher Rensing

Point-by-point response to the comments from the reviewers

Reviewer #1:

The revised manuscript is much better now. The manuscript still has multiple format errors.

Response: We appreciate the reviewer’s comments. According to your suggestion, we have gone over the article carefully and revised the manuscript.

  1. Tables 1 and 2, the first column needs a heading. Table 2 should have a similar layout as in Table 1 (i.e., the first column should be the independent variable = the test bacteria). Standardize the format of Table legends.

Response: Thanks for the comment. We have considered the change in Tables 1 and 2.

  1. Figure 2. The lettering is too small and illegible. Magnifying the text to a legible size makes it blurred.

Response: Thanks for the comment. We have replaced Figure 2 with a readable figure.

  1. Figure 3. Explain the color codes? The headline GBDP phylogeny based on 16S data should read 16S gene sequence data.

Response: Thanks for the comment. We have revised Figure 3 and added an explanation of color codes below it.

  1. Avoid using the ampersand sign in the text.

Response: Thanks for the comment. We have replaced all the “ampersand”.

  1. Mixed format of upper-case and lower-case letters in Table 4.

Response: Thanks for the comment. We have replaced all the letters in Table 4 with lower-case.

  1. References. Mixed format of upper case and lower letters in article titles. Incorrect journal title abbreviations. Missing volume numbers and article IDs. For chapters in books, include the editor and publisher. Check the proper use of the subscript and superscript letters.

Response: Thanks for the comment. We have carefully reformatted the references in accordance with journal requirements.

  1. Note p-hydroxybenzoic acid, hydroxyisobutyric acid, glutathione must not be capitalized!

Response: Thanks for the comment. We have made the relevant modification as requested.